# FUNCTIONAL RELATION FIELD: A MODEL-AGNOSTIC FRAMEWORK FOR MULTIVARIATE TIME SERIES FORECASTING

## ABSTRACT

In multivariate time series forecasting, the most popular strategy for modeling the relationship between multiple time series is the construction of graph, where each time series is represented as a node and related nodes are connected by edges, i.e. spatial-temporal graph neural networks. The graph structure is either given *apriori* or learned based the similarity between nodes. However, the relationship between multiple time series is typically complicated, for instance, the sum of outflows from upstream nodes may be equal to the inflows of downstream nodes. Such relations widely exist in many real-world multivariate time series forecasting scenarios, yet are far from well studied. In these cases, graph might only be a crude description on the dependency between nodes. To this end, we explore a new framework to model the inter-node relationship in a more precise way based our proposed inductive bias for graphs, *Functional Relation Field*, where a group of functions parameterized by neural networks are learned to characterize the dependency between multiple time series. These learned functions are versatile: they can then be used to discover the underlying graph structure by identifying the most relevant neighbors of the target node; and on the other hand, the learned functions will form a "field" where the nodes in the backbone prediction networks are enforced to satisfy the constraints defined by these functions. The experiment is conducted on one toy dataset to show our approach can well recover the true constraint relationship between nodes. And two real-world MiniApp calling traffic and road networks datasets are also considered with various different backbone networks. Results show that the prediction error can be reduced remarkably with the aid of the proposed functional relation field framework.

## 1 INTRODUCTION

Multivariate time series forecasting has surged recently due to its strong expressiveness of the spatio-temporal dependence among the data and its enormous popularity in vast application areas, such as the prediction of urban traffic, computer network flow, cloud micro-services calling flow, and rigid body motion, to name a few (Li et al., 2018; Yu et al., 2018; Bai et al., 2020; Yan et al., 2018; Liu et al., 2020). The most popular and straightforward strategy for modeling the relationship between multiple time series is the introduction of graph, where each time series is represented as a node and related nodes are connected by edges. This particular inductive

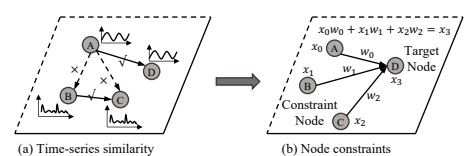

(a) Time-series similarity    (b) Node constraints

Figure 1: Comparison between traditional graph-based modeling and our approach. (**a**) Graph structure learning with time-series similarity; (**b**) Functional relation field, modeling inter-node functional relationship in a linear form, where $D$ is the target node and $\{A, B, C\}$ represent dependent nodes.

bias for multivariate time series prediction results in the so called spatial-temporal graph neural networks (Yu et al., 2018). The graph structure is either given *apriori* (e.g. in traffic flow prediction, each road as a node has connected roads forming the graph.) or learned based the similarity between nodes (Yu et al., 2019; Bai et al., 2020; Shang et al., 2021). However, in practice, the relationship

between multiple time series is typically complicated. For instance, there often exist constraints among the nodes, ranging from the equality between the inflow and the outflow for a node in a traffic network to the geometric constraints of the rigid body motion. Such relations widely exist in many real-world multivariate time series forecasting scenarios, yet are far from well studied. In these cases, graph might not be sufficient for characterizing the dependency between nodes.

As a remedy, in this work, we explore a new framework to model the inter-node relationship in a more precise manner than graph, **Functional Relation Field** (FRF), where a group of functions parameterized by neural networks are learned to characterize the dependency between multiple time series explicitly. These learned functions are versatile: first they can then be used to discover the underlying graph structure by identifying the most relevant neighbors of the target node; and on the other hand, the learned functions will form a "field" where the nodes in the backbone prediction networks are further enforced to satisfy the constraints defined by these functions. As illustrated in Fig.1, the left panel shows the traditional graph neural networks assuming similar time series have edge connections, while our framework on the right panel models the dependency between nodes through *a functional relationship*, e.g. a linear form to enforce the constraints between the flows of target and dependent nodes. In our framework, we mainly solve the following two issues: (i) *How to learn the functional field?* We need to select the dependent nodes that have a relationship with the target node, and express the constraint in a functional form; (ii) *How to guarantee the constraints satisfaction?* The (functional) constraints relationship should be maintained in the predicted output in both training and test process.

To address these issues, we propose a two-stage approach that can discover the functional relations (i.e. constraints) from data and further integrate the constraints seamlessly when forecasting the multivariate time series. Specifically, we first train a neural network with a selected target node as its output and all the other nodes as dependent variables (i.e. the input of this neural network), and identify the most relevant dependent nodes based on this trained network. We then re-train it to learn the relationship among the target and the discovered relevant nodes. Next, we incorporate these functional constraints into the network backbones by imposing them to the predicted output during both training and test process. More precisely, the output of the network could be guaranteed to satisfy the constraints by utilizing the constraint-satisfied transformation and loss minimization. We compare the proposed approach with SVM, fully connected networks, fully connected LSTM, and five backbone models (i.e., STGCN (Yu et al., 2018), AGCRN (Bai et al., 2020), Autoformer (Wu et al., 2021), FEDformer (Zhou et al., 2022), SCINet (Liu et al., 2022)). Experimental results show that our approach significantly improves the performance over the original network backbones and other baseline models.

## RELATED WORK

**Univariate time series forecasting.** Recently, much research focuses on time series forecasting with deep learning models due to their powerful representational capability and prediction performance, including feed-forward neural network, RNN (Rumelhart, 1986) and its variants LSTM (Hochreiter & Schmidhuber, 1997) and GRU (Cho et al., 2014). The transformer architecture and its variants (Vaswani et al., 2017; Simm et al., 2020; Zhou et al., 2021; Child et al., 2019; Lim et al., 2020; Li et al., 2019; Wu et al., 2021; Zhou et al., 2022) also made much progress on univariate time-series forecasting on learning long-range dependence. In order to model the trend and seasonality of time series in an interpretable way, N-beats (Oreshkin et al., 2020) network that stacked very deep full-connection network based on backward and forward residual links has improved the multi-horizon prediction accuracy significantly. Moreover, DeepAR (Salinas et al., 2020) and Deep State-Space Model (DSSM) (Rangapuram et al., 2018) stack multi-layer LSTM network to generate parameters of one-step-ahead Gaussian predictive distributions for multi-horizon prediction.

**Multivariate time series forecasting.** Spatio-temporal graph neural networks (Yu et al., 2018; Chen et al., 2019; Pan et al., 2021; Li et al., 2020) have been proposed to model the spatial correlation and temporal dependency in multivariate time-series. Apart from capturing the temporal dependence, these methods further model the spatial dependence among all time series via graph neural networks, leveraging the information from the neighboring time series to help forecasting the target one. It is well known that an informative graph structure is important to the graph time series forecasting. Therefore, many algorithms (Bai et al., 2020; Seo et al., 2016; Shang et al., 2021) were proposed to discovery the underlying graph structure. AGCRN (Bai et al., 2020) assumed the graph structure

is unknown and adopted an adaptive approach to learn the embedding vectors for all nodes, and then replaced the adjacency matrix in graph convolutions with a function of the node embeddings. However, the similarity graph calculated with the learned node embedding is a dense and continuous graph instead of a sparse and discrete graph. Therefore, GTS (Shang et al., 2021) formulated the graph structure learning problem as a probabilistic graph model to learn the discrete graph through optimizing the mean performance over the graph distribution.

Different from the existing multivariate time series prediction methods, AGCRN (Bai et al., 2020) (with a fully connected graph) and STGCN (Yu et al., 2018) (with a given graph), we consider a more precise way, i.e. functional relations as constraints, to learn the connection between time series. The new inductive bias expressed by these functional relations can be applied to different backbone networks to help recover the graph structure and act as regularization in both training and test process.

## 2 METHODOLOGY: FUNCTIONAL RELATION FIELD

**Multivariate time series forecasting.** Suppose we have $N$ time series $\{x_i\}_{i=1}^N$ with length $T$, written compactly as $X \in R^{N \times T}$. Each time series can be denoted as a node, where $x_{i,t} \in R$ for each node $i$ and time step $t$. $x_t \in R^N$ is the time slice of $X$ at the $t$-th time step. The multi-step forecasting problem of a multivariate time series can be formulated as predicting the future $M$ frames of the multivariates given the last $H$ time slices:

$$\{\hat{y}_{t+1}, ..., \hat{y}_{t+M}\} = \operatorname{argmax} P(\{y_{t+1}, ..., y_{t+M}\}|\{x_{t-H+1}, ..., x_t\}), \tag{1}$$

where $\{y_{t+1}, \cdots, y_{t+M}\}$ and $\{\hat{y}_{t+1}, \cdots, \hat{y}_{t+M}\}$ represent the true and predicted values at the future time steps, $M$ is the number of future steps. Note that here we use $y$ to denote the output so as to differentiate it from the input $x$.

**Forecasting with functional relations.** In many real-world scenarios, the relationship between multiple time series is typically complicate, graph might not be sufficient for modelling their dependency, particularly for the cases values of multivariate time series at each time step are subject to some *intrinsic constraints*. Existing methods have not incorporated these constraints into their models. In this work, we intend to show that models with the account of constraints (expressed with functional relationship) are superior to those without constraints in terms of prediction performance. As an example, suppose that the flow in a computer network satisfies the homogeneous linear constraints, at each time step $t$, the following linear constraints hold for slice $x_t$:

$$Ax_t = 0, \forall t, \tag{2}$$

where $A \in R^{M \times N}$ is a matrix that is constant across time. In other more complex cases, the constraints can be non-homogeneous, non-linear, or even intertemporal. Here, we concentrate on time-invariant constraints that is not intertemporal. As such, the constraints can be described by a set of functions $f$ with size $m$, i.e. functional relation field,

$$f = (f_1, f_2, ..., f_m). \quad f_i(x_t) = 0, \ \forall i, \ \forall t. \tag{3}$$

Based on the constraints defined above, we consider the following constrained multivariate time series prediction problem,

$$\begin{aligned} \{\hat{y}_{t+1}, ..., \hat{y}_{t+M}\} &= \arg\max P(\{y_{t+1}, ..., y_{t+M}\}|\{x_{t-H+1}, ..., x_t\}), \\ s.t. \quad f_i(\hat{y}_{t+\tau}) &= 0, \quad 1 \le \tau \le M, \quad 1 \le i \le m. \end{aligned} \tag{4}$$

However, in most real-world scenarios, neither the functional form $\mathcal{F}$ nor the specific weights variables involved in the constraints are given, and one of our objectives is to extract such information from the data and solve the problem (4). We now elaborate the functional relation field for multivariate times series prediction in the following.

The schematic diagram of the proposed framework is depicted in Figure 2, including two parts. The first part displayed Figure 2(a) shows how we learn the functional relations, i.e. the constraints between nodes. Assuming that the constraints are unknown, we aim to find the constrained nodes and the specific functional form for these constraints. The constraint function in this paper is

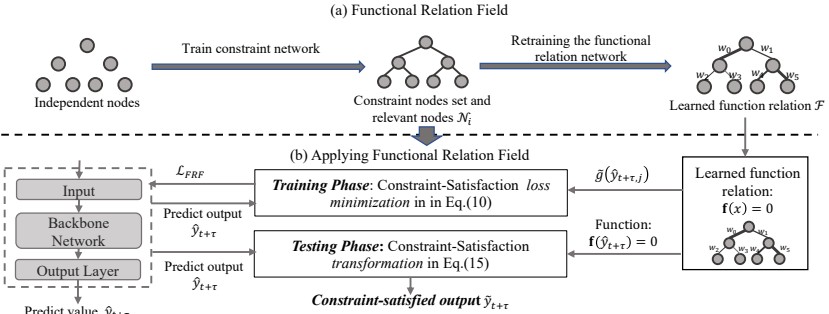

Figure 2: The schematic diagram of functional relation field framework. The two subfigures denote the two stages: (**a**) The training data is employed to discover the nodes in each constraint function and these functions are expressed by constraint network; (**b**) The learned constraints are incorporated in the backbone models (cf. Section 2.2) in three complementary ways so as to improve the forecasting performance.

approximated by a neural network, named as functional relation network or constraint network. After training the functional relation network, we can identify the most relevant neighbors and produce a more informative graph structure. Then we can proceed to integrate the learned constraints into the backbone graph neural networks for multivariate time series prediction, as shown in Figure 2(b). We enforce these constraints to the output of spatio-temporal graph neural networks during both training and test phases. For the outputs of the networks, we add a constraint-satisfied transformation layer during the inference process such that the outputs strictly satisfy the constraints. Altogether, we refer to the proposed framework as functional relation field-enhanced spatio-temporal graph networks (FRF-STG). It is model-agnostic and can be applied to different backbone graph networks. In the following, we will describe the two stages including learning functional relation network and how to apply the constraints induced by the functional relation between nodes in more details.

## 2.1 LEARNING THE FUNCTIONAL RELATION NETWORK

We start with discussing the first question: *how to learn the unknown constraints (i.e. the functional relations) from the multivariate time series data?* As demonstrated in Figure 2(a), we assume that there exists a constraint for each node. We first discover the relevant nodes involved in these constraints and then express the constraint functions via neural networks.

**Identifying constrained nodes and their relevant nodes.** Here we consider a simplified case where the functional relation between nodes can be formulated as:

$$x_{t,i} = g_i(x_{t,\setminus i}), \forall t \tag{5}$$

i.e. for each target node $i$, we use a constraint network $g_i$ to approximate the function relation taking all the remaining $(N-1)$ nodes as input. We then train the constraint network to predict the value of the $i$-th node with the loss function :

$$\mathcal{L}_{pred,(i)} = \|\hat{x}_{t,i} - x_{t,i}\|^2 \tag{6}$$

where $\hat{x}_{t,i}$ and $x_{t,i}$ represent the estimated and observed values of node $i$ at time step $t$. Second, a threshold $\epsilon_{err}$ is set, and treat $x_i$ as a constrained node if both the training and validation error are smaller than $\epsilon_{err}$. Otherwise, $x_i$ is unpredictable with the other nodes, indicating it has weak dependency with other nodes. Then, to identify the most relevant nodes set $\mathcal{N}_i$ for target node $i$, we introduce the sensitivity of input change to the output for the trained constraint network, measured by the absolute value of the partial derivative:

$$\delta_{i,j} = \left| \frac{\partial g}{\partial x_{t,j}} \right|, j \neq i \tag{7}$$

We calculate the average gradients over the training and the validation set for node $j$. Then, we specify another threshold $\epsilon_{grad}$ here and consider the node $j$ as the most relevant node of target $i$ if $\delta_{i,j}$ is larger than $\epsilon_{grad}$. Besides, if the cardinality of $\mathcal{N}_i$ is larger than the scale threshold $J$, we further shrink $\mathcal{N}_i$ by only keeping the top-$J$ nodes with the largest $\delta_{i,j}$.

**Retraining the functional relation network.** Since we filter out the irrelevant nodes for the discovered constrained node $x_i$, it is necessary to re-train the constraint network using the relevant nodes in $\mathcal{N}_i$ as inputs, denoted as $x_{t,\mathcal{N}_i} = \{x_{t,ij} | j \in \mathcal{N}_i\}$,

$$\hat{x}_{t,i} = \tilde{g}_i(x_{t,\mathcal{N}_i}). \tag{8}$$

Regarding the architecture of the functional relation network $\tilde{g}_i$, we adopt a simple attention-based structure for each node $i$, described as follows.

$$\alpha_{t,i} = Softmax(MLP_i(x_{t,\mathcal{N}_i})), \quad \hat{x}_{t,i} = \alpha_{t,i}^T x_{t,\mathcal{N}_i}, \tag{9}$$

where $\alpha_{t,i}$ is the attention weight vector generated from the relevant nodes $x_{t,\mathcal{N}_i}$, and $\hat{x}_{t,i}$ is the reconstructed input with the constraint nodes. Others alternatives for designing the functional relation network is also possible.

## 2.2 Applying the Constraints

The constraints learned by the functional relation network are versatile. A naive usage is to construct meaningful graph structure by drawing edges between the identified target and its dependent nodes. Secondly, we propose to incorporate the learned constraints into the backbone prediction network in both training and test process through *constraint-satisfaction loss minimization* and *constraint-satisfaction transformation*, respectively. Both of them are used to guarantee that the constraints are maintained in the outputs of the backbone network.

**Constraint satisfaction in training phase.** We expect the output of the backbone network, $\hat{y} = \{\hat{y}_{t+1}, \hat{y}_{t+2}..., \hat{y}_{t+M}\}$, to satisfy the learned constraints that could reveal the underlying structure of the multivariate time series. A straightforward yet effective way of implementing the constraint satisfaction is loss minimization over the functional relation network based on the output of the backbone prediction network,

$$\mathcal{L}_{FRF}(\hat{y}) = \sum_{i=1}^{N} \sum_{\tau=1}^{M} \|\hat{y}_{t+\tau,i} - \tilde{g}(\{\hat{y}_{t+\tau,j}\}, j \in \mathcal{N}_i)\|_2^2 \tag{10}$$

Therefore, the overall loss function for training the backbone prediction network include two terms,

$$\mathcal{L}_{total} = \mathcal{L}(\hat{y}, y) + \lambda \mathcal{L}_{FRF}(\hat{y}), \tag{11}$$

where $\lambda$ is a tradeoff coefficient for balancing the supervised term and constraint satisfaction.

**Constraint satisfaction in testing phase.** Furthermore, although the constraints are fully utilized during training, there is no guarantee that the constraints hold for the outputs during the inference process. Therefore, it is necessary to perform *constraint-satisfaction transformation* on outputs of the prediction networks.

Let us first consider the linear constraint $Ax_t = 0, \forall t$. Suppose that $\hat{y} = \{\hat{y}_{t+1}, \hat{y}_{t+2}..., \hat{y}_{t+M}\}$ and $y = \{y_{t+1}, y_{t+2}, ..., y_{t+M}\}$ denote the predicted output of the backbone network and the ground truth, respectively. To make the output $\hat{y}_{t+\tau}$ to satisfy the linear constraint, we can project the predicted output onto the hyperplane $Ax_t = 0$ as $\tilde{y}_{t+\tau}$ with a closed-form solution,

$$\tilde{y}_{t+\tau} = \hat{y}_{t+\tau} - A^T(AA^T)^{-1}A\hat{y}_{t+\tau}. \tag{12}$$

On the other hand, for non-linear constraint set $f(y) = (f_1(y), ..., f_m(y))^T = 0$, where each constraint $f_i(y) = 0$ represents $y_i - \tilde{g}_i(y_{t,\mathcal{N}_i}) = 0$, there are no analytical solutions, but we can solve an optimization problem with nonlinear equality constraints, i.e. finding the nearest projection point on the plane $f(y) = 0$ given the reference point $\hat{y}_{t+\tau}$ for $\tau = 1, \ldots, m$

$$\min_{\tilde{y}_{t+\tau}} \|\tilde{y}_{t+\tau} - \hat{y}_{t+\tau}\|_2^2, \text{ s.t. } f(\tilde{y}_{t+\tau}) = 0. \tag{13}$$

A simple approximate method for solving this equality-constrained quadratic programming is to conduct iterative projections. Denote $\mathcal{J} = \frac{\partial f}{\partial x}$ as the Jacobian matrix. Assuming $\hat{y}_{t+\tau} \approx \tilde{y}_{t+\tau}$, closed to the surface $f(x) = 0$. We derive the first-order Taylor expansion of $f(x)$ at $\hat{y}_{t+\tau}$ as

$$f(x) \approx f(\hat{y}_{t+\tau}) + \mathcal{J}^T \cdot (x - \hat{y}_{t+\tau}). \tag{14}$$

Equating $f(x)$ to zero with $x = \tilde{y}_{t+\tau}$ yields

$$\tilde{y}_{t+\tau} = \hat{y}_{t+\tau} - \mathcal{J}(\mathcal{J}^T\mathcal{J})^{-1}f(\hat{y}_{t+\tau}). \tag{15}$$

Then we can repeat the above transformation several times (e.g. number of projections $K = 10$ times used in our experiments) until the constraints are well satisfied by evaluating whether $F(x) = \sum_{j=1}^{m} |f_j(x)|$ is small enough.

## 2.3 Functional Relation Field-enhanced Spatio-Temporal Graph Networks

In this part, we integrate the proposed functional relation field framework into five representative backbone models, STGCN (Yu et al., 2018), AGCRN (Bai et al., 2020), Autoformer (Wu et al., 2021), FEDformer (Zhou et al., 2022) and SCINet (Liu et al., 2022) to boost their prediction performance, referred as FRF-STGCN, FRF-AGCRN, FRF-Autoformer, FRF-FEDformer and FRF-SCINet, respectively. In the first stage, we learn the functional relation network, based on which the most relevant nodes can be identified. And the resultant graph structure could be used for the five backbone networks. In the second stage, we enforce the learned constraints in the training and inference process, as described in Figure 2.

Since different backbone networks has their own specific design, we need adapt FRF to these backbones. For the constraint satisfaction of output, in AGCRN and SCINet, the networks produce all the prediction results at multiple time steps in one batch, and therefore, the constraint-satisfied transformation is applied to the prediction at each time step respectively for $K$ times as described in Eq. (15). For STGCN, we apply the above transformation sequentially to each future time step, obtain the transformed predictions, and then feed the predictions to STGCN to produce the predictions at the next time step. We repeat this procedure until we finish the multi-step forecasting task.

---

**Algorithm 1:** Training and inference of functional relation field

---
**Input:** Trained function relation networks $f$, hyper-parameters $\lambda$ and $K$.
**Output:** constraint-satisfied output $\tilde{y}_{t+\tau}$
`// Training Phase;`
  **repeat**
1     |   Forward on backbone network to get $\hat{y}_{t+\tau}$               ▷ on training dataset;
2     |   Back-propagate with the loss $\mathcal{L}_{total}$ in Eq. 2.2 and run Adam.       ▷ constraint-satisfaction loss
3 **until** *stopping criteria is met*;
  `// Inference Phase;`
  Forward on the trained backbone network to obtain $\hat{y}_{t+\tau}$            ▷ on test dataset;
4 **for** *k in K* **do**
5     |   Calculate $\tilde{y}_{t+\tau}$ by Eq.(15)       ▷ constraint-satisfaction transformation;
6 **end**

---

## 3 Experiment

In this section, we conduct experiments on five datasets including one synthetic graph dataset, two real-word MiniApp calling flow datasets and two traffic flow datasets to demonstrate the effectiveness of FRF on learning the underlying relationship between nodes and boosting prediction performance of these backbone networks. The code for reproducibility is attached in the Supplementary Materials.

**The baseline models.** We first compare our framework with two traditional forecasting models including Historical Average (HA) and Support Vector Regression (SVR). Then, we also conduct experiments on two classical univariate time series prediction models, including Feed-Forward Neural Network (FNN) and Full-Connected LSTM (FC-LSTM (Sutskever et al., 2014)). We select the widely used graph time series model STGCN (Yu et al., 2018), AGCRN (Bai et al., 2020), and the univariate time series forecasting models based on transformer architectures Autoformer (Wu et al., 2021), FEDformer (Zhou et al., 2022) and another state-of-the-art univariate prediction model SCINet (Liu et al., 2022)) as our backbone networks. We refer the readers to the supplementary materials for the detailed experimental settings.

### 3.1 Datasets and settings

**Binary tree dataset.** We first generate an artificial graph time series dataset. The graph structure for this dataset is a complete binary tree with 255 nodes. For each leaf node $i$, its value is a noisy sinusoidal wave across time, $x_{i,t} = n_{i,t} A_i \sin(\frac{2\pi t}{T_i} + \phi)$, where $n_{i,t} \sim \mathcal{U}(0.95, 1.05)$. We sort all leaf nodes from left to right in an increasing order of their periods. For a non-leaf node $p$, we denote its left and right child as $l$ and $r$. We further set the value of node $p$ to be the geometric mean of its two children $l$ and $r$, $x_{p,t} = \sqrt{x_{l,t} \cdot x_{r,t}}$. We sample one point every 5 minutes, so there are 288 points per day. We generate the data for 40 days, including 30 days for training (i.e., $30 \times 288 = 8640$ time points), 5 days for validation, and 5 days for testing. We intentionally design this dataset since it has true graph structure between different time series and the constraints between nodes are explicit,

and thus it is a suitable testbed to compare the superiority of FRF over those without FRF. In the experiments, for the backbone with FRF, we assume the constraints are unknown and learn them using the proposed method in Section 2.1.

**MiniApp calling flow dataset 1 and 2.** These two datasets are real-word flow data from two popular online payment MiniApps, attached in the Supplementary Materials. For the two MiniApps, there are $N = 30, 23$ filtered pages linking to each other in the calling process, which produces visiting request flow from one page to another, constituting a graph with $N = 30, 23$ nodes. We aggregate the flow with averaged value every 5 minutes for each node, so there are 288 points per day. For the first MiniApp, we collect 21 days of data, including 15 days for training, 3 days for validation, and 3 days for test. For the second one, 24 days of data are collected, including 18 days for training, 3 days for validation, and 3 days for testing.

**PEMSD4 and PEMSD8 traffic datasets.** This benchmark dataset is popular for multi-variate time series prediction, describing the traffic speed in San Francisco Bay Area with 307 sensors on 29 roads (`https://paperswithcode.com/dataset/pemsd4`). The other one consists of 170 detectors on 8 roads in San Bernardino area (`https://paperswithcode.com/dataset/pemsd8`).

**Settings of constraint network and hyper-parameters.** For the architectures of the constraint network, we compare two a 4-layer MLP and a self-attention network, and the results show the latter is more effective. We measure the constraint relationship with MAPE, where the large MAPE indicates the time-invariate constraint is weak. Specifically, the MAPEs for BinaryTree, MiniAPP1, MiniApp2, PEMSD4, PEMSD8 datasets are 0.10, 0.008, 0.01, 0.02, 0.07 respectively. The larger MAPE means the weaker constraint relationship, therefore the proposed FRF model is applicable to backbone network only when the MAPE of constraint network is small. In addition, we only tune the parameters of FRF while keeping the other hyper-parameters setting the same as backbone networks.

## 3.2 RESULTS

**Overall performance** Table 1 summarizes the performance of all the compared models on the five datasets, including the proposed FRF approach coupled with STGCN, AGCRN, Autoformer, FED-former and SCINet, denoted as FRF-STGCN and FRF-AGCRN, FRF-Autoformer, FRF-FEDformer and FRF-SCINet, respectively.

For the binary tree dataset, we predict the future 12 time steps and evaluate the performance in terms of three metrics (MAE, RMSE, MAPE). Since the underlying true constraints are known, we report the experimental results of our models with both true and learned constraints, denoted as "T" and "L". We can observe that deep learning-based models typically outperform the traditional ones, as expected. Furthermore, the proposed functional relation field can further improve the performance of the original backbone models. Regardless of the differences between the two backbone networks, FRF can consistently improve the prediction accuracy for both of the backbones. indicating that the FRF framework could be potentially applied to a wide variety of backbones.

For the two MiniApp datasets, we omit the metric MAPE since the scale of data changes dramatically across time such that MAPE fails to characterize the performance of different models. Due to the error accumulation problem for multi-step prediction in STGCN, the performance of this model pales in comparison with its non-iterative counterpart. As a result, we only report the results of the non-iterative version of STGCN. Since the underlying true constraint relationship between nodes are not available, we only report the FRF with learned constraints. We can easily observe that augmentation of the proposed FRF can consistently boost the performance of the five backbone networks. Specifically, FRF improves STGCN by 36.3% and 6.9% on the two datasets, also improves AGCRN by 14.6% and 7.0%, respectively.

For traffic datasets PEMSD4 and PEMSD8, one particular reason we choose SCINet as the baseline is that the reported results can achieve state-of-the-art prediction performance on this task. We can observe that even relying on such a strong baseline, FRF framework can still improve its performance of with a margin 0.6% and 0.3% on both datasets, respectively. For other backbones, we again see that FRF further improves the prediction performance, showing the effectiveness of FRF as a model-agnostic framework.

**Learning the relationship between nodes.** We further test whether FRF could discover the underlying true constraints between nodes. First, we investigate whether we can reliably estimate the

Table 1: Model performance on BinaryTree and MiniApp datasets. "(T)" and "(L)" represent the models with true and learned constraints, respectively. Bold font is used to show the advantage over backbones. "-" represents that the ground truth of the functional relationship is not available.

| Methods | Binary tree | | | MiniApp 1 | | MiniApp 2 | |
|---|---|---|---|---|---|---|---|
| | MAE | RMSE | MAPE | MAE | RMSE | MAE | RMSE |
| HA | 12.64 | 19.19 | 22.16% | 3.97 | 9.77 | 11.02 | 35.23 |
| SVR | 8.71 | 14.00 | 15.85% | 2.56 | 7.06 | 6.83 | 21.68 |
| FNN | 5.77 | 10.04 | 9.73% | 2.09 | 6.26 | 5.43 | 16.84 |
| FC-LSTM | 17.08 | 22.83 | 32.40% | 2.05 | 4.08 | 8.14 | 19.64 |
| STGCN | 2.65 | 5.82 | 4.36% | 1.90 | 5.26 | 4.50 | 14.14 |
| FRF-STGCN (T) | **2.40** | **5.68** | **3.94%** | - | - | - | - |
| FRF-STGCN (L) | 2.50 | 5.71 | 4.12% | **1.21** | **3.32** | **4.19** | **11.11** |
| AGCRN | 2.56 | 5.77 | 4.25% | 0.41 | 1.17 | 1.43 | 3.79 |
| FRF-AGCRN (T) | **2.30** | **5.54** | **3.87%** | - | - | - | - |
| FRF-AGCRN (L) | 2.37 | 5.57 | 4.00% | **0.35** | **0.92** | **1.33** | **3.39** |
| Autoformer | 8.54 | 13.16 | 15.41% | 1.03 | 2.79 | 2.69 | 6.85 |
| FRF-Autoformer (T) | **8.34** | **12.79** | **14.75%** | - | - | - | - |
| FRF-Autoformer (L) | **8.34** | 12.83 | 14.38% | **0.77** | **2.18** | **2.46** | **5.70** |
| FEDformer | 8.54 | 13.24 | 15.45% | 0.60 | 1.80 | 2.08 | 5.13 |
| FRF-FEDformer (T) | **8.10** | **12.80** | **14.40%** | - | - | - | - |
| FRF-FEDformer (L) | 8.29 | 12.99 | 14.79% | **0.58** | **1.76** | **2.03** | **4.98** |
| SCINet | 5.43 | 9.37 | 9.46% | 0.52 | 1.51 | 1.78 | 3.88 |
| FRF-SCINet (T) | **5.36** | 9.34 | 9.53% | - | - | - | - |
| FRF-SCINet (L) | 5.37 | **9.27** | **9.43%** | **0.47** | **1.34** | **1.71** | **3.65** |

Table 2: Model performance on two traffic datasets, PeMSD4 and PeMSD8. "-" represents the value is too large and thus ignored.

| Model Type | Methods | PEMSD4 | | | PEMSD8 | | |
|---|---|---|---|---|---|---|---|
| | | MAE | RMSE | MAPE | MAE | RMSE | MAPE |
| Multivariate | STGCN (Yu et al., 2018) | 21.61 | 35.25 | 13.84% | 17.28 | 27.19 | 11.14% |
| | FRF-STGCN | **20.70** | **33.90** | **13.46%** | **16.46** | **26.05** | **10.68%** |
| | AGCRN(Bai et al., 2020) | 19.81 | 32.58 | 13.18% | 16.52 | 26.12 | 10.53% |
| | FRF-AGCRN | **19.59** | **31.85** | **13.08%** | **16.04** | **25.28** | **10.30%** |
| Univariate | Autoformer | 21.42 | 34.09 | - | 18.49 | 28.78 | - |
| | FRF-Autoformer | **21.24** | **33.93** | - | **18.23** | **28.66** | - |
| | FEDFormer | 21.59 | 34.23 | - | 18.52 | 29.23 | - |
| | FRF-FEDFormer | **21.29** | **33.82** | - | **18.15** | **28.61** | - |
| | SCINet(Liu et al., 2022) | 19.27 | 31.27 | 11.91% | 15.71 | 24.60 | 10% |
| | FRF-SCINet | **19.15** | **31.09** | **11.80%** | **15.67** | **24.57** | **10%** |

Figure 3: Performance comparison of three kinds of hyper-parameters including $\epsilon_{err}$, $\lambda_R$ and $K$ on binary tree dataset with the SOTA backbone AGCRN.

target node given the values of constraint nodes. To be exact, we compute $\hat{x}_{t,i} = \tilde{g}(\{x_{t,\mathcal{N}_i}\})$ and compare $\hat{x}_{t,i}$ with $x_{t,i}$ in terms of MAPE. For the test data of the synthetic binary tree, the resulting MAPE is $0.399\%$. Note that the MAPE of AGCRN or STGCN reported in Table 1 is around $4\%$ without considering the constraints. Therefore, using the learned constraints can well regularize the predictions given by the original network backbones as well as further improve the forecasting performance. On the other hand, we compare the performance of the proposed algorithm when using the true and estimated constraints, showing the results in Table 1. We can observe that the performance based on both the true and estimated constraints is almost the same, indicating that the constraints are accurately learned. Additionally, we visualize the learned constraints by connecting each constrained node with their most relevant neighbors as a graph, shown in Figure 4. The structure of the binary tree is well recovered, although some extra edges are involved.

**Hyperparameters Sensitivity.** FRF enhanced model introduces additional three kinds of hyper-parameters including validation error threshold $\epsilon_{err}$, the loss tradeoff coefficient $\lambda$ and the number of output transformation $K$. Therefore, we conduct hyper-parameters sensitivity experiments on binary

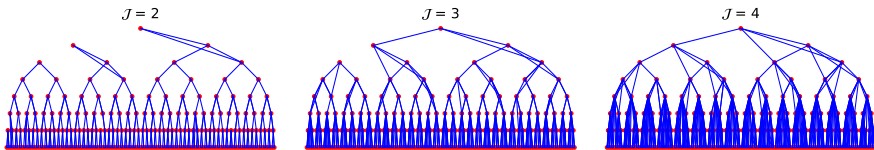

Figure 4: The learned constraints of the Binary Tree Dataset by connecting each constrained node with their most related nodes. We use $\mathcal{J} = 2, 3, 4$ for every node to plot this figure, so there are lack of connections when $\mathcal{J} = 2$ and some redundant connections when $\mathcal{J} = 4$.

Table 3: Ablation study on explicit graph, constraint graph learned from constraint network and constraint satisfaction components using STGCN as the backbone network.

| # | Explicit Graph | Constraint Graph | Constraint Satisfaction | PeMSD4 MAE | PeMSD4 RMSE | PeMSD8 MAE | PeMSD8 RMSE | MiniApp1 MAE | MiniApp1 RMSE |
|---|---|---|---|---|---|---|---|---|---|
| 1 | ✓ | | | 21.61 | 35.25 | 17.28 | 27.19 | 1.90 | 5.26 |
| 2 | | ✓ | | 21.26 | 35.13 | 16.79 | 26.61 | 1.48 | 3.87 |
| 3 | | ✓ | ✓ | 20.70 | 33.90 | 16.46 | 26.05 | 1.21 | 3.32 |

Table 4: Ablation study on constraint-satisfaction loss minimization and constraint-satisfaction transformation. The backbone AGCRN is used for Binary Tree, MiniApp1 and MiniApp2; and SCINet for PEMSD4.

| # | FRF Training | FRF Inference | Binary Tree MAE | Binary Tree RMSE | MiniApp1 MAE | MiniApp1 RMSE | MiniApp2 MAE | MiniApp2 RMSE | PEMSD4 MAE | PEMSD4 RMSE |
|---|---|---|---|---|---|---|---|---|---|---|
| 1 | ✗ | ✗ | 2.56 | 5.77 | 0.41 | 1.17 | 1.43 | 3.79 | 19.27 | 31.27 |
| 2 | ✓ | ✗ | 2.51 | 5.68 | 0.40 | 1.19 | 1.38 | 3.66 | 19.20 | 31.16 |
| 3 | ✗ | ✓ | 2.33 | 5.58 | 0.40 | 1.14 | 1.45 | 3.74 | 19.22 | 31.15 |
| 4 | ✓ | ✓ | 2.30 | 5.54 | 0.35 | 0.92 | 1.33 | 3.29 | 19.15 | 31.09 |

tree dataset using backbone AGCRN as shown in Fig 3. We can observe that the performance slightly improves when the $\epsilon_{err}$ increases due to more constraints are discovered, while the performance decreases with large $\epsilon_{err}$ because of the introduced noise. Even more, the FRF enhanced model performs worse than backbone network when $\epsilon_{err} = 5.0$. Consistently, FRF enhanced model performs better when $\lambda = 0.1$ and worse than backbone with large $\lambda$. For the $K$, the larger $K$ improves the backbone more significantly than smaller $k$ because iterating more times makes the non-linear constraint optimization problem more accurate.

**Ablation Study.** We first conduct an ablation study on the constraint graph learned from constraint network using the STGCN as backbone network in Table 3. We can observe that the constraint graph performs better than explicit graph extracted from prior knowledge on both traffic and MiniApp datases. In addition, for backbone networks without explicit graph structure such as AGCRN and SCINet, we investigate the effectiveness of constraint-satisfaction loss minimization and constraint-satisfaction transformation as shown in Table 4, finding that both of the two components contribute to the forecasting performance. Specifically, for the backbone network AGCRN which achieves the state-of-the-art performance on binary tree dataset, FRF enhances the backbone by $1.95\%$ in training phase and by $9.0\%$ in inference phase, while the combination of two components improves the performance by $10.16\%$ in total.

# 4 CONCLUSION

In this paper, we have proposed to enhance the multivariate time series forecasting with a new inductive bias, function relation fieild (FRF), which is model-agnostic. FRF can discover the intrinsic graph structure, as well as improve flow forecasting performance by applying constraint function relationship to the output in training and testing phases. The constraints learned by FRF can be incorporated into existing backbone networks, consistently improving the prediction performance. Experimental results show that the proposed FRF framework can reliably learn the constraints from the time-series data and restore the graph structure. Moreover, these constraints in turn help improve the prediction accuracy by a notable margin, regardless of the diversity of the network architecture in different backbone models. We expect that this FRF inductive bias could be potentially employed in other multivariate settings beyond times series scenarios.

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

## A    PERFORMANCES ON MORE BACKBONES

**GTS** Shang et al. (2021). The discrete graph structure learning model learns a graph structure among multiple time series and forecasts them simultaneously with a GNN. There are two differences between GTS and our proposed FRF. On one hand, GTS performs prediction under GNN paradigm which is model-specific while FRF is model-agnostic applying the function field to forecasting loss optimization. On the other hand, existing studies including AGCRN and GTS construct the graph based on the time-series similarity, while the FRF is the first proposed to exploiting the the constraint function relation to enhance the multi-variate time-series forecasting. We conduct experiments on Binary tree, Miniapp1 and Miniapp2 datasets using the opensource code (`https://github.com/chaoshangcs/GTS.git`) shown in table.5, demonstrating that FRF can also improve the forecasting performance on GTS. *The code of FRF-GTS and the running log is released in the supplementary material.*

**NRI** Kipf et al. (2018). The neural relational inference (NRI) model is an unsupervised model that learns to infer interactions and forecasting with a lstm. We conduct experiments on Binary tree, Miniapp1 and Miniapp2 dataset using the opensource code (`https://github.com/ethanfetaya/NRI.git`). The results on NRI network in table.5 showing that there is a large margin from the SOTA backbone AGCRN Bai et al. (2020).

Table 5: Performance comparison of FRF enhanced GTS and NRI Networks on BinaryTree and MiniApp datasets. FRF-AGCRN is the SOTA model.

| Methods | Binary tree | | | MiniApp 1 | | MiniApp 2 | |
|---|---|---|---|---|---|---|---|
| | MAE | RMSE | MAPE | MAE | RMSE | MAE | RMSE |
| NRI Kipf et al. (2018) | 22.77 | 30.15 | 39.58% | 2.50 | 6.89 | 8.04 | 16.92 |
| GTS Shang et al. (2021) | 5.85 | 9.19 | 8.87% | 1.92 | 2.32 | 3.88 | 7.26 |
| FRF-GTS (T) | **5.67** | **8.21** | **7.85%** | - | - | - | - |
| FRF-GTS (L) | 5.70 | 8.33 | 8.08% | **1.77** | **1.84** | **2.70** | **5.14** |
| FRF-AGCRN ( SOTA ) | **2.30** | **5.54** | **3.87%** | **0.35** | **0.92** | **1.33** | **3.39** |

## B    EXPERIMENTAL SETTINGS

**The error threshold.** For the binary tree dataset and MiniApp calling flow datasets which have strong constraint relationships, we set $\epsilon_{err} = 0.01$ to filter the constaint nodes. However, for traffic dataset PEMSD4 and PEMSD8 with relative weak constraints, we set $\epsilon_{err} = 0.025$ to achieve the best performance. The hyper-parameters sensitivity experiments of $\epsilon_{err}$ on PEMSD4 and PEMSD8 datasets are shown in Fig 5.

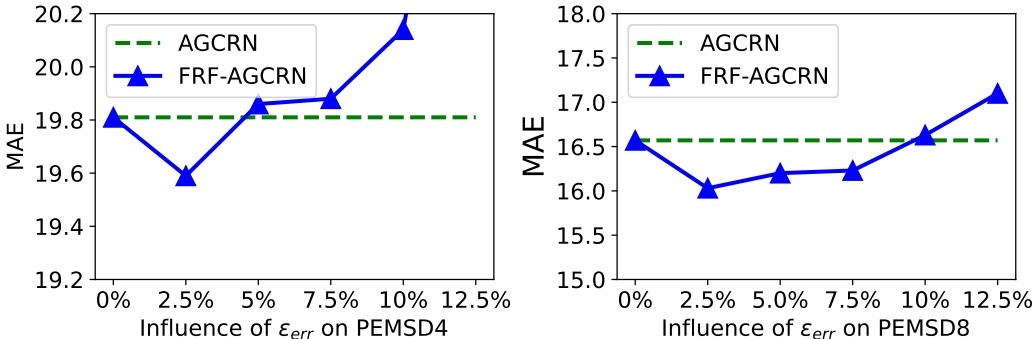

Figure 5: Performance comparison of $\epsilon_{err}$ on PEMSD4 and PEMSD8 dataest on backbone AGCRN.

**The function relation graph.** Note that for the real datasets, the graph structure is not given in advance. In order to use STGCN, we adopt Gaussian copula graphical models Liu et al. (2009); Yu et al. (2020) to learn the graph structure from the data, and take the learned graph as benchmark graph. For the FRF enhanced backbone network STGCN Yu et al. (2018), we replace the fixed graph structure with the learned constraint graph then achieve better performance. As results shown in table

3, we can observe that constraint graph performs better than graph learned with copula graphical model.

Besides, for uni-variate backbones SCINet, Autoformer and FEDformer taking no time-series relationship into consideration, As well as graph model AGCRN, which have optimized with learned node embedding dynamically ignoring the origin graph, we don't exploit constraint relation at graph construction stage. The function relation is applied in training stage and output constraints.

**The setting of $J$.** For binary tree dataset, we set $J = 4$ to recover the function relation shown in Fig 4. We set $J = 6$ for two MiniApp flow calling datasets. For traffic dataset PEMSD4 with 307 nodes and PEMSD8 with 170 nodes, we achieve best performance when $J = 30$.

**The detailed settings at $\lambda$ and $k$.** In the training stage, we only tune the trade off coefficient $\lambda$ and iteration times $K$ while keep all other parameters the same as SOTA settings in benchmark. The detailed settings are shown in 6.

Table 6: Detailed hyper-parameter settings of all graph time-series and univariate backbone networks on five datasets.

| Methods | Binary tree | | MiniApp 1 | | MiniApp 2 | | PEMSD4 | | PEMSD8 | |
|---|---|---|---|---|---|---|---|---|---|---|
| | $\lambda$ | $K$ | $\lambda$ | $K$ | $\lambda$ | $K$ | $\lambda$ | $K$ | $\lambda$ | $K$ |
| FRF-STGCN | 0.1 | 10 | 0.01 | 5 | 0.01 | 10 | 0.01 | 5 | 0.001 | 10 |
| FRF-AGCRN | 0.1 | 10 | 0.01 | 5 | 0.01 | 10 | 0.01 | 5 | 0.1 | 10 |
| FRF-Autoformer | 0.01 | 20 | 0.001 | 5 | 0.001 | 5 | 0.001 | 5 | 0.001 | 10 |
| FRF-FEDformer | 0.01 | 20 | 0.001 | 5 | 0.001 | 5 | 0.001 | 5 | 0.001 | 10 |
| FRF-SCINet | 0.01 | 10 | 0.001 | 5 | 0.001 | 5 | 0.0001 | 5 | 0.0001 | 5 |

## C  VISUALIZATION OF LEARNED FUNCTION RELATION

**The flow visualization of different relations.** We show the comparison of learned function relation and origin relation on MiniApp1 dataset in table 6. Note that, the origin relation of MiniApp is learned by Gaussian copula graphical models Liu et al. (2009); Yu et al. (2020). We can observe that the flows of the target node has the same pattern and scale with relevant node on learned function, while has different scale on origin graph. The results demonstrating that learned function is more effective to capture the flow relationship.

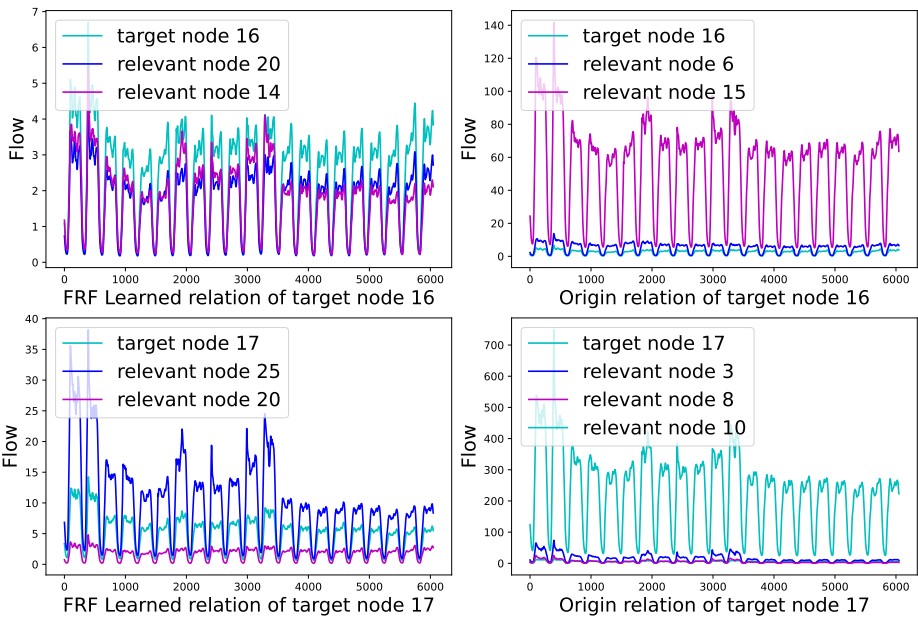

Figure 6: Flow visualization of learned function and origin relation on MiniApp1 dataset.

# D    DISCUSSION ON HYPERPARAMETERS AND COMPUTATIONAL COMPLEXITY

**Hyper-parameters.** There are three newly-introduced hyper-parameters including error threshold $\epsilon_{err}$, trade-off coefficient $\lambda$ and number of iterations $K$. The $\epsilon_{err}$ and $\lambda$ can be easily chosen based on the validation loss. And a larger $K$ could be used to obtain more accurate optimization and achieve better performance. So, there is a balance between performance gain and computation. We typically set it as $K = 10$ which could work well for all the tasks we have considered.

**Computational complexity.** On one hand, the computational complexity increases in the forecasting network training caused by the $K$ iterations of output constraint satisfaction. The $K$ is usually setted as a small number 5 or 10, which is computationally easy. And the main time-consuming operations come from forward and back propagation of backbones rather than the output constraint. On the other hand, we need to train the constraint network for all time-series. Fortunately, the constraint network is a simple two-layer attention network, which only has a small number of parameters but effective enough to capture the complex function relation. For example, in MiniApp1 task, each constraint network only has around 3,000 parameters, the training time is in the scale of seconds. Thus, we believe training a constraint network is very fast and does not require much computational resources. The small size of the constraint networks is amenable to a large-scale multi-variate time series.

