# OpenReview forum: "Functional Relation Field: A Model-Agnostic Framework for Multivariate Time Series Forecasting"
_ICLR.cc/2023/Conference — Submitted to ICLR 2023_

### Official Review · Reviewer_3BB6 · 2022-10-24

**Confidence:** 3
**Correctness:** 3
**Technical Novelty And Significance:** 3
**Empirical Novelty And Significance:** 3
**Recommendation:** 5

**Clarity, Quality, Novelty And Reproducibility:**

The motivation is strong, the proposed method is novel. But the writing is hard to follow, some concepts are not well-explained.

**Strength And Weaknesses:**

Strength:
1. The motivation for exploring relationships between nodes in terms of constraints is insightful.

Weaknesses:
1. The writing is hard to follow.
2. Some parts are not clear:
(1) what is the physical meaning of linear constraints (equ. 2)?
(2) In equ. 3, why set f_i(x_t)=0
(3) In section 2.2, is the target-dependent relationship bidiractional?
(4) It seems the proposed model needs to go over all the nodes to check the target and dependent, what is the complexity of the method?
3. \epsilon_{grad} is a hyperparameter. But the experiment does not evaluate the effects of \epsilon_{grad}.
4. It seems too many hyperparameters to control the constraint learning. It may expect much more effort for hyperparameter tuning, which may jeopardize the applicability of the method.
5. Typo: the second paragraph on page 2, "...while our framework on the left panel models the..." it should be" the right panel"?

**Summary Of The Paper:**

This paper proposed a method to automatically learn constraints over nodes in a graph, and apply the constraints for spatial-temporal forecasting. The major contribution is the proposed constraint learning framework.

**Summary Of The Review:**

The motivation is strong, the proposed method is novel. But the writing is hard to follow, some concepts are not well-explained.

---

> ### Author Response · Authors · 2022-11-16
> **Thank you for recognizing the novelty of our approach.**
>
> Thank you very much for recognizing the novelty and strong motivation of our approach. About the clarity issue and details we will answer accordingly.  In the following, we take "Q" as the question from reviewers and "A" as our answer.
>
> About the issue "The writing is hard to follow."
>
> **Q1: "Some parts are not clear: (1) what is the physical meaning of linear constraints (equ. 2)?"**
>
> A1: The linear constraint could be interpreted as at time t, one variable can be linearly represented by other variables. In some real-world applications, linear constraints could model the inter-node relationship. For example, the sum of upstream flows may be equal to its downstream flow in web or miniapp calling applications.
>
> **Q2: "(2) In equ. 3, why set f_i(x_t)=0"**
>
> A2: The equation of $f_i(x_t)=0$ represents the function relation between target node and constraint nodes, which is an equality constraint.
>
>
> **Q3: "(3) In section 2.2, is the target-dependent relationship bidirectional?"**
>
> A3: We are not sure about what this question exactly means. For each node, we train a small neural network to identify whether this node can be a target node and which nodes can be its dependent nodes. From this perspective, the target-dependent relationship is indeed bidirectional.
>
> **Q4: "(4) It seems the proposed model needs to go over all the nodes to check the target and dependent, what is the complexity of the method?"**
>
> A4: Yes. It is true that we need to go over all the nodes to learn the constraint relation. Fortunately, the constraint network is a small two-layer attention network, which has a small number of parameters but effective enough to capture the inter-node relation according to our experiments. For example, in MiniApp1 task, each constraint network only has only around 3,000 parameters, the training time is in the scale of seconds. Thus, we believe training a constraint network is very fast and does not require much calculation resources.  As shown in Figure 4, the binary tree structure is fully discovered by a simple constraint network. We will add more discussions related to this issue in the updated version of the paper.
>
>
> **Q5: "\epsilon_{grad} is a hyperparameter. But the experiment does not evaluate the effects of \epsilon_{grad}."**
>
> A5: Thank for pointing out this issue. We conduct experiments on $\epsilon_{grad}$ varying from [0.1, 0.2, 0.3 ,0.4, 0.5] on miniapp2 dataset under FRF-AGCRN, and the mae changed in the range [1.33, 1.40, 1.42, 1.47, 1.55]. Although the prediction error increases with larger $\epsilon_{grad}$, the frf enhanced model performs better than bakcbone (mae 1.43 on AGCRN) in most cases, demonstrating frf is insensitive to $\epsilon_{grad}$. We will release the code and running log in the future.
>
>
> **Q6: "It seems too many hyperparameters to control the constraint learning. It may expect much more effort for hyperparameter tuning, which may jeopardize the applicability of the method."**
>
> A6: Since the main goal of our FRF is to improve the backbone networks, thus we set the hyperparameters of the backbones same as the original and only tune the newly introduced three hyperparameters $\epsilon_{err}$, $\lambda$, $K$. These hyperparameters can be easily tuned by the validation loss. Additionally, as we emphasized in A4, the constraint network is quite small and thus its hyperparameters are computationally easy to tune. Lastly, in Figure 3, we test the sensitivity of the hyperparameters, showing that FRF could perform better than the backbones on a large range of hyperparameter settigs when the constraint relation is well learned.
>
>
> **Q7: Typo: the second paragraph on page 2, "...while our framework on the left panel models the..." it should be" the right panel"?**
>
> A7: Thanks for your reminding, and we have fixed this typo in the revised version.

---

### Official Review · Reviewer_n4W8 · 2022-10-26

**Confidence:** 5
**Correctness:** 3
**Technical Novelty And Significance:** 4
**Empirical Novelty And Significance:** 3
**Recommendation:** 6

**Clarity, Quality, Novelty And Reproducibility:**

Clearly and well written, organized and explained - with some grammar and typos that should be fixed (see below).  I find the approach to be highly novel as mentioned, and reproducibility is good as code is provided and details around experiments, datasets, and hyper parameters are reported.  I do feel the complete experiment process / how all hyper parameters and architectures are chosen is not fully explained.


Grammar issues and typos throughout hurt readability - e.g., in intro: "...learned based the similarity..." instead of "...learned based on the similarity...",
"...a more precise manner than graph...", "...the introduction of graph...", "...were proposed to discovery the...", "...relationship between multiple time series is typically complicate...", "Others alternatives ... is also possible", (typo) "function relation fieild"


Incorrect statement: "finding the nearest projection point on the plane f(y) = 0" if f is nonlinear as described, this may not define a plane


**Strength And Weaknesses:**

Strengths:

1) I find the approach to be interesting and novel - I personally have not seen this kind of approach taken before, and I feel it can be seen as generalizing a couple common classes of approaches: hierarchical forecasting - which does incorporate functional relationships but of a fixed form (hierarchical aggregates), and graph-based forecasting - which does not specify the particular relationship between nodes as is done here.  This enhances the hierarchical approach by generalizing the functional form encoded to be not just hierarchical or linear, and also generalizes the graph approach to encode more specific relationships between variables explicitly.

2) The paper was well-written and organized so it was easy to follow and understand.

3) The method is logical and sound.

4) The number of datasets and different models compared to and enhanced with the proposed method lend good credence to the proposed approach, along with the additional analyses - ablation study, hyperparameter sensitivity, and analysis of the relationships learned.


Weaknesses:

1) A graph learning / graph based approach (e.g., with graph neural nets) could theoretically learn arbitrary relationships between nodes as well (as for example this would be encoded in multiple layers of graph neural networks) - the authors never really say what is the advantage of the proposed approach compared to this idea.

2) The learned relationships are static / stationary - that is for the same series values x, the relationship will be the same regardless of time.  This seems a bit over-restricting and not realistic for time series which often have non-stationary relationships as well.
It seems like this approach might be over-constraining to enforce some relationship learned across all time points - i.e., in many cases it may be more realistic if the functional relationship changes over time or context (which could also be captured by encoding context in additional exogenous series).

3) Additionally, some prior work on learning the graph structure for forecasting allows the graph structure learned to be influenced by the forecast modeling as well - which is a disadvantage of this approach as it is learned separately.

4) It would have been useful to also see a comparison with prior work on reconciliation (at least given a known structure) - as typical approaches can encode arbitrary linear relationships between time series as well

5) As the constraints are learned (in a completely uncontrolled fashion with flexible neural nets) - how can you guarantee the constraints don't contradict each other at all?  If they do, how can the constraint projection work and what do you do in those cases?  This could easily happen for example if the model learns some (incorrect) relationship like: x_1 = 2 * x_2 and x_2 = 2 * x_1 for 2 variables / time series x_1 and x_2 (as a simplified case to illustrate the point).

6) A major weakness is the large number of hyper parameters introduced by the approach - beyond the hyper parameters used for each forecast model itself as well.  There are even more than really tested and pointed out because the relationship network architecture and training introduces its own set of hyper parameters as well (and this could even be done for the two different types of networks for learning the relationship as well).  Several hyper parameters are chosen seemingly without the typical validation process (seemingly choosing whatever gave the best results) - along with the fact of being so many that need to be precisely tuned this limits practical usefulness of the method and confidence it will work well on other datasets for real use (where we have to do our best to select hyper parameters for everything).

Applying this method, the number of hyper parameters that need to be tuned is quite daunting - tuning the hyper parameters (architecture and learning) for the relationship networks.  Tuning the various relationship thresholds.  Tuning lambda and K in the learning objective.  And finally tuning the multiple hyper parameters of the backbone forecast model as well.  It would be helpful for the authors to add some discussion on how to address this complexity and for clear description of how all hyper parameters were chosen (some hyper parameter numbers are just reported in tables and mentioned they worked better, but the process for choosing them is not clearly explained).  Discussing this issue and how it could be addressed, and further study if certain fixed values or procedures would work sufficiently, could strengthen this work.


7) Another major weakness of the proposed methods is the limited scalability - and lack of discussion around this weakness along with lack of analysis of computational complexity / reported run times as a function of the number of time series.    Adding these could help strengthen this work.
In particular the computational complexity of the method seems daunting, as we have to train a neural network with the roughly the same amount of data as for the forecast model itself, for every single time series, twice, just for learning the functional relationships.  So for thousands of time series this amounts to training thousands of neural networks, which can be further multiplied for hyper parameter tuning / optimization.
Some discussion should really be added around this, and also if there are any ways to address it, and would strengthen the paper.


8) Also repeated experiments are not performed - and std. dev. of metric scores / confidence intervals are not reported so it's hard to determine the significance of differences in metric scores, and robustness of the results.  Ideally metric scores would be averaged over multiple random runs and multiple test time series windows.

**Summary Of The Paper:**

The authors propose a new model for multivariate forecasting with structured relationship between time series.  They propose to learn a static functional relationship between each node and every other node (where the target node/series value is equal to a function of the values of the other nodes/series), and then incorporate these functional relationships into any existing forecast model - via adding a regularization term (penalizing forecasts that deviate from the relationships) to the regular forecast loss (weighted by a hyper parameter lambda), and also via subsequently projecting the final forecasts onto the (learned) constraints.

The authors validate their approach by comparing results on several datasets for several recent forecast methods with and without the proposed functional relationship component and demonstrate consistently improved forecast error metrics using the proposed approach.  They also perform a variety of hyper parameter sensitivity and ablation analysis, and some analysis around learned functional relationships.

**Summary Of The Review:**

Overall I found the approach to be interesting and novel and could be seen as a generalization / enhancement of two classes of approaches: hierarchical forecasting and graph-based forecasting.  For the former it can be seen as extending the functional relationship to more general kinds (rather than just hierarchical aggregates) and for the latter, enabling encoding more information about the relationships between series.

Potential weaknesses include not learning the relationships end to end - so they can not be influenced by the downstream forecast modeling, and the functional relationships being too restricting and non-dynamic (as they are fixed over time).  However the latter can be controlled to some extent with hyper parameters (such as lambda), and both may be beneficial to avoid overfitting.

The major weaknesses that prevent me from raising my score higher mainly arise because of the large number of hyper parameters introduced (not just the ones mentioned but even the hyper parameters associated with the relationship network architecture and training) along with lack of clear, principled hyper parameter selection and no repeated randomized experiments to provide confidence intervals / std. dev., and the limited scalability of the method.  The latter of which is not discussed (and no run times reported).  In particular it's hard to imagine this method could be used well in practice with so many hyper parameters that must be tuned, and it would really only be tractable for a small number of time series as for n time series, 2n networks have to be trained just for the relationship learning part of the model.

---

> ### Author Response · Authors · 2022-11-15
> **Thank for your recognition on the novelty and solidness of our work and the reply is as follows - Part 3**
>
>
> **Q8: "Another major weakness of the proposed methods is the limited scalability...Some discussion should really be added around this, and also if there are any ways to address it, and would strengthen the paper."**
>
> A8: Thanks for the valuable comments and I will add the discussion on computational complexity to the updated version.
> On one hand, the computational complexity increases in the forecasting network training caused by the $K$ times iteration of output constraint satisfaction. $K$ is usually setting to a small number $5$ or $10$, which is computationally easy. And the main time-consuming operations come from forward and back propagation of backbones rather than the output constraint. On the other hand, it is true that we need to train constraint network for all time-series. Fortunately, as mentioned in A5, the constraint network is a simple two-layer attention network, which only has a small number of parameters but effective enough to capture the complex function relation. Thus, the small size of the constraint networks is amenable to a large-scale multi-variate time series.
>
>
> **Q9: "Also repeated experiments are not performed - and std. dev. of metric scores / confidence intervals are not reported so it's hard to determine the significance of differences in metric scores, and robustness of the results. Ideally metric scores would be averaged over multiple random runs and multiple test time series windows."**
>
> A9: In the experiments, we fix all the random seed the same as backbone setting for fair comparison, since we want to show after incorporating FRF it could improve over the backbone. Therefore the experimental results can are reproducible and repeat multiple times will obtain the same results.
>
>
> **Q10:
> "Potential weaknesses include not learning the relationships end to end - so they can not be influenced by the downstream forecast modeling, and the functional relationships being too restricting and non-dynamic (as they are fixed over time). However the latter can be controlled to some extent with hyper parameters (such as lambda), and both may be beneficial to avoid overfitting."**
>
> A10: We would like to hightlight the introduced FRF framework is **model-agnostic** that it can be applied to arbitary time series prediction models. We first learn the functional relationship between nodes. This could be thought as an inductive bias that will be incorporated to training and test phases of the backbone network in the form of regularization. So the idea is to let FRF influce forescasting network, not vice versa. Although there are some works [1][2] that attempt to learn the relation and forecasting end-to-end, they are all model-specific under fixed paradigm such as STGCN rather than model-agnostic.
> About the stationarity of the constraints, in the current version we assume they are static that are already sufficient for many applications at least for considered ones in our paper. For instance, the miniapp calling relationship rarely changes over time, almost one or two years. For traffic flow dataset, the the road network and traffic condition is also unchanged given a fixed network topology.
>
> [1] Wu, Zonghan, et al. "Connecting the dots: Multivariate time series forecasting with graph neural networks." Proceedings of the 26th ACM SIGKDD international conference on knowledge discovery & data mining. 2020.
>
> [2] Li, Zhuoling, et al. "Dynamic Graph Learning-Neural Network for Multivariate Time Series Modeling." arXiv preprint arXiv:2112.03273 (2021).

---

> ### Author Response · Authors · 2022-11-15
> **Thank for your recognition on the novelty and solidness of our work and the reply is as follows - Part2**
>
>
> **Q4: "It would have been useful to also see a comparison with prior work on reconciliation (at least given a known structure) - as typical approaches can encode arbitrary linear relationships between time series as well. As the constraints are learned (in a completely uncontrolled fashion with flexible neural nets) - how can you guarantee the constraints don't contradict each other at all? If they do, how can the constraint projection work and what do you do in those cases? This could easily happen for example if the model learns some (incorrect) relationship like: x_1 = 2 * x_2 and x_2 = 2 * x_1 for 2 variables / time series x_1 and x_2 (as a simplified case to illustrate the point)."**
>
> A4: In this paper, we assume the relationship is static and could be reflected by real-world data observations. We learn the constraint relation network by optimizing the loss to **a very small value even approach zero** when the constraint relation actually exists. If existing $x_1=2 * x_2$ and $x_2=2* x_1$, then the loss of either of them must be large. Therefore, we think constraint could not contradict with each other when the network is well trained and the error is strictly controlled.
>
>
> **Q5: "A major weakness is the large number of hyper parameters introduced by the approach - beyond the hyper parameters used for each forecast model itself as well..."**
>
> A5: There are three newly-introduced hyper-parameters including error threshold $\epsilon_{err}$, trade-off coefficient $\lambda$ and number of iterations $K$.  $\epsilon_{err}$ and $\lambda$ can be easily chosen based on the validation loss. And a larger $K$ could be used to obtain more accurate optimization and achieve better performance. So, there is a balance between performance gain and computation. We typically set it as K=10 which works well.
> More importantly, the constraint network is a small two-layer attention network, which has a small number of parameters but effective enough to capture the inter-node relation according to our experiments. For example, in MiniApp1 task, each constraint network only has only around 3,000 parameters, the training time is in the scale of seconds. Thus, we believe training a constraint network is very fast and does not require much computational resources.
>
>
> **Q6: "Several hyper parameters are chosen seemingly without the typical validation process (seemingly choosing whatever gave the best results) - along with the fact of being so many that need to be precisely tuned this limits practical usefulness of the method and confidence it will work well on other datasets for real use (where we have to do our best to select hyper parameters for everything)."**
>
> A6: It is true that the hyper-parameters are precisely tuned on traffic dataset PEMSD4 and PEMSD8. The reason is that the constraint relation is not sufficiently obvious due to  the noise in traffic data. In the experiments, a large validation loss in constraint network training process indicates the function relation is weak. Therefore, the function relation is measurable and observable. However, for the fully learned constraint function relation in binary tree and Miniapp dataset, the performance is insensitive to the hyper-parameters. As shown in Figure 3, FRF performs better than backbones with a large range of $\epsilon_{err}$, $\lambda \in [0.001, 0.1]$ as well as $K \in [5, 10, 15, 20, 25]$, demonstrating that there is no need to precisely tune the hyper-parameters to get performance gain when the constraint relation is precise.
>
>
> **Q7: "Applying this method, the number of hyper parameters that need to be tuned is quite daunting - tuning the hyper parameters (architecture and learning) for the relationship networks. ..."**
>
> A7: Thanks for this valuable comments. We fix the hyper-parameters of the backbone because it already performs very well on public setting. And thus after incorporating FRF into the backbone prediction network, we only tune the three newly-introduced hyper-parameters of the constraint network, i.e. the error threshold $\epsilon_{err}$, the coefficient $\lambda$ and the iteration times $K$. Additionally, with the precisely learned constraint function such as binary tree and minapp dataset, the performance is not very sensitive to all these three hyper-parameters as shown in Figure 3. The FRF performs better than the original backbones on large range of hyper-parameter settings.

---

> ### Author Response · Authors · 2022-11-15
> **Thank for your recognition on the novelty and solidness of our work and the reply is as follows - Part1**
>
> Thank for your recognition on the novelty and solidness of our work. In the following, we will answer your questions regarding the details of the paper accordingly.  We take "Q" as the question from reviewers and "A" as our answer.
>
> **Q1: "A graph learning / graph based approach (e.g., with graph neural nets) could theoretically learn arbitrary relationships between nodes as well (as for example this would be encoded in multiple layers of graph neural networks) - the authors never really say what is the advantage of the proposed approach compared to this idea."**
>
> A1: It is true that graph neural network can learn arbitrary relationship between nodes. However, compared with traditional multi-layer GNN, one of the most important contribution of FRF is to learn an **explicit** functional relationship between nodes such that it can be directly applied to forecasting loss optimization. And our framework is **model-agnostic** and could be **directly adpated** to various types of model architectures. We used a two-layer attention network to learn the function relation, which is simple, explicit yet efficient. There are many existing works attempt to learn the node relationship with graph neural network such as paper [1] and paper [2]. However, paper [1] is under the STGCN paradigm, which is model-specific rather than model-agnostic. Thank you for pointing out this, we will add more discussions to the updated version of the paper.
>
>
> **Q2: "The learned relationships are static / stationary - that is for the same series values x, the relationship will be the same regardless of time. This seems a bit over-restricting and not realistic for time series which often have non-stationary relationships as well."**
>
> A2: It is true that dynamic relationship may be better than static relationship. However, there are two considerations when choosing the stationary relationship. On one hand, in the considered applications, the relationship might be stationary in reality. For instance,  the miniapp calling relationship rarely changes over time, almost one or two years. For traffic flow dataset, the road network and traffic condition are also unchanged given a fixed network topology. Therefore, we think the stationary relationship is realistic and efficient for some real-world forecasting tasks, at least for those considered in our paper. On the another hand, the non-linear optimization has no analytical solution. A simple approximate method for solving this problem is to conduct iterative projections. It is extremely challenging to solve the variable-constrained programming problem. The dynamic relationship modelling can be taken as an important point for future exploration.
>
>
> **Q3: "Additionally, some prior work on learning the graph structure for forecasting allows the graph structure learned to be influenced by the forecast modeling as well - which is a disadvantage of this approach as it is learned separately."**
>
> A3: It is a good question that there are many models learning the graph structure and forecasting in an end-to-end way. We consider this question from the following three aspects. First, the graph structure learned by our constraint network is effective to reflect the groundtruth relationship, so it may be not necessary to learn the relation end-to-end. As shown in Figure 4, the binary tree structure is fully discovered by a simple 2-layer attention network. Second, solving of non-linear constraied optimization problem has no analytical solution and requires iterative procedures, so it is difficult to optimized together with forecasting. A possible solution could be that we first fix the constraint network optimization as a quadratic loss, then fix the loss and optimize the constraint networks. We will leave this as future work. But it is hard to ensure there are obvious performance gain when learning relationship with together forecasting. Third, existing models learning graph structure and forecasting simultaneously in papers [1,2] is specific to certain models such as STGCN, AGCRN, while our proposed paradigm is model-agnostic which can also be used in univariate time-series forecasting models such as autoformer, fedformer, scinet, as well as multi-variate graph models.

---

### Official Review · Reviewer_RLjD · 2022-11-04

**Confidence:** 4
**Correctness:** 3
**Technical Novelty And Significance:** 2
**Empirical Novelty And Significance:** 2
**Recommendation:** 3

**Clarity, Quality, Novelty And Reproducibility:**


In this paper, some formulation is not precise and clear enough, and the presentation needs improvements as listed above.

The problem studied in this paper is not new. The technical contribution seems marginal.



**Strength And Weaknesses:**

Strength:

This paper explores the idea of capturing hidden inter-variate constraints/relations in multivariate time series and imposing the discovered constraints on the training and inference process.

In the experiments, the proposed framework is applied to baselines to compare the performance difference. This is a valid idea.

Weakness:

The problem of identifying inter-variate constraints/relations and applying them in forecasting is not new, and several works have studied it, e.g., [1, 2]. The technical contribution in this paper looks marginal, given that the technique used in this paper is standard and no news insights seem uncovered. Meanwhile, the presentation, some design choices, and evaluation have the following issues.

(a) Eq.(1) seems to formulate the time series forecasting from the perspective of the probability model, while the rest of the paper follows the standard point forecast paradigm. In the probability model, the forecast is not necessarily the mode and could be mean, quantile, or interval. Eq.(1) seems disconnected from the problem of this paper.

(b) Eq.(4)-(6) present the "constraint", which is not rigorous w.r.t. the concept of constraint. The presented constraint is essentially closer to the concept of correlation or relation, since it is simply derived from how well the other variables fit the target variable.      This is highly data or observation-dependent. Moreover, it needs a threshold to determine the set of relevant variables. For multi-variables in different value domains or distributions, finding proper thresholds seems nontrivial and would affect the overall performance.      This way of identifying  "constraint"  seems ad-hoc and arbitrary.

(c) From Eq.(10), the constraint discovered from X seems to be applied to Y. It is a bit confusing to present this way.

(d) Eq.13 - Eq.15 seems problematic. In Eq.(13), the minimization and the constraint are mutually exclusive, i.e., the minimization problem is for relaxing the constraint, and if the constraint is to behold, the minimization is unnecessary. Meanwhile, Eq.(13) is simply a  least-squares problem w.r.t. $\tilde{y}$ and the iterative process seems redundant.

(e) In the experiment, only the newly introduced hyperparameter is compared. It would be good to also show the hyperparameters in training, since they affect the end performance significantly in many cases.

[1] Wu, Zonghan, et al. "Connecting the dots: Multivariate time series forecasting with graph neural networks." Proceedings of the 26th ACM SIGKDD international conference on knowledge discovery & data mining. 2020.

[2] Li, Zhuoling, et al. "Dynamic Graph Learning-Neural Network for Multivariate Time Series Modeling." arXiv preprint arXiv:2112.03273 (2021).





**Summary Of The Paper:**

The paper focuses on the problem of time series forecasting with constraints.  The proposed functional relation field framework is aimed at learning constraints from multi-variate time series data. Then, the authors develop the training and inference method incorporating the learned constraints. The proposed method is evaluated on both synthetic and real datasets.

**Summary Of The Review:**

The paper focuses on the problem of learning hidden constraints in multivariate time series and making use of the learned constraints in the training and inference of forecasting models.

The problem is not new. The technical contribution is incremental, given that some design choices are standard, and some seem problematic and need more clarification.

Some experiment results look interesting, however, given the aforementioned weakness, it needs improvements to be more solid and convincing.

---

> ### Author Response · Authors · 2022-11-15
> **Thank for your important comments and we would like to answer these questions-Part2**
>
> In the following, we take "Q" as the question from reviewers and "A" as our answer.
>
> **Q6: "This way of identifying "constraint" seems ad-hoc and arbitrary."**
>
> A6: Although the way of identifying "constraint" is simple, the two-layer attention network is effective to capture the complex constraint functional relationship. Demonstrated in Figure 4, the binary tree structure is fully recovered with the simple constraint network. In addition, the learned constraint function improves all backbone networks, which also shows that the discovered function relation is sufficiently accurate. Therefore, the way of identifying "constraint" is efficient and effective in most real-world cases.
>
>
>
> **Q7: "From Eq.(10), the constraint discovered from X seems to be applied to Y. It is a bit confusing to present this way."**
>
> A7: As stated in the paper, we denote X as the historical observed data ${x_1, ..., x_t}$, Y as the future data ${x_{t+1}, ..., x_{t+\tau}}$ to be predicted. The X and Y are the same time series in different time intervals, so they should have the same constraint function relation. This is the reason why we applied the discovered constraints from X to Y.
>
>
> **Q8: "Eq.13 - Eq.15 seems problematic. In Eq.(13), the minimization and the constraint are mutually exclusive, i.e., the minimization problem is for relaxing the constraint, and if the constraint is to behold, the minimization is unnecessary."**
>
> A8: The proposed FRF is a standard optimization problem with a non-linear constraint. Simply, the optimization we need to solve is to minimize the forecasting loss under the equality constraint. More detailed could be referred to https://en.wikipedia.org/wiki/Constrained_optimization.
>
> **Q9: "Meanwhile, Eq.(13) is simply a  least-squares problem w.r.t. and the iterative process seems redundant."**
>
> A9: The least-square problem we need to solve is a standard nonlinear optimzation problem with constraints. Since it has no analytical solution, it need to be solved iteratively which is a standard nonlinear optimization procedure.
>
> **Q10: "In the experiment, only the newly introduced hyperparameter is compared. It would be good to also show the hyperparameters in training, since they affect the end performance significantly in many cases."**
>
> A10: For a fair comparison, we keep hyperparameters of backbone networks unchanged. This is because the hyperparameters showed in origin paper for the backbone models such as STGCN, AGCRN and SCINet are already well tuned and achieved SOTA performance in many datasets. So we think we should follow the same settings as the backbones to check whether our newly proposed FRF can improve backbone or not. Remarkably, after incorporating FRF with only newly introduced hyperparameters, all the models achieved better performance than the original backbones, demonstrating the effectiveness and generality of modeling functional relation field.
>
> [1] Wu, Zonghan, et al. "Connecting the dots: Multivariate time series forecasting with graph neural networks." Proceedings of the 26th ACM SIGKDD international conference on knowledge discovery & data mining. 2020.
>
> [2] Li, Zhuoling, et al. "Dynamic Graph Learning-Neural Network for Multivariate Time Series Modeling." arXiv preprint arXiv:2112.03273 (2021).
>
> [3] Bing Yu. "Spatio-Temporal Graph Convolutional Networks: A Deep Learning Framework for Traffic Forecasting" IJCAI 2018.
>
> [4] Syama Sundar Rangapuram. "Deep State Space Models for Time Series Forecasting". NeurIPS 2018

---

> ### Author Response · Authors · 2022-11-15
> **Thank for your important comments and we would like to answer these questions-Part1**
>
> In the following, we take "Q" as the question from reviewers and "A" as our answer.
>
> **Q1: "The problem of identifying inter-variate constraints/relations and applying them in forecasting is not new, and several works have studied it, e.g., [1, 2]."**
>
> A1: We now compare our FRF framework with [1,2] to show that it is a novel perspective to model the inter-variate relationship.
> (1) The paper [1] learned the relationship with GNN to capture spatial correlation, and used temporal convolution to capture inner-variate relation then optimized both parts iteratively. However, the approach in [1] is only applicable to specific backbone architecture, e.g. STGCN. Thus, it is not general enough to be applied to backbones models such as transformer, SCINet and AGCRN. The model in paper [1] is **model-specific** rather than **model-agnostic** network, compared to our proposed FRF framework.
>
> (2) The paper [2] learned the static and dynamic graph structure with node embedding and changing node features, based on the assumption that similar node features may have similar patterns. The most significant difference from [2] is that our FRF proposed a functional field with the assumption that there are functional relation between input flow and outflow. Indeed, exploiting the similarity to generate graph structure is not new, which is widely studied in many works. However, we are the first to generate the graph structure from  perspective of constraint function rather than similarity. Additionally and more importantly, the constraints between nodes described by functions (rather than (un)weighted edges) is more precise and general can be applicable to many backbones including STGCN, SCINet, transformer and AGCRN.
>
> As a summary,  to the best of our knowledge, FRF  is the first model-agnostic framework to solve the multi-variate time-series forecasting with functional relation field.
>
>
> **Q2: "Eq.(1) seems to formulate the time series forecasting from the perspective of the probability model, while the rest of the paper follows the standard point forecast paradigm. In the probability model, the forecast is not necessarily the mode and could be mean, quantile, or interval. Eq.(1) seems disconnected from the problem of this paper."**
>
> A2: The formulation of time series forecasting with probability model is widely used in existing forecasting models [3][4]. For example, both the Eq.1 in paper [3] STGCN and the Eq.1 in paper [4] Deep State Space Models formulate the forecasting as a  probability model. All these models could be optimized with maximum likelihood estimation to learn the parameters.
>
>
> **Q3: "Eq.(4)-(6) present the "constraint", which is not rigorous w.r.t. the concept of constraint. The presented constraint is essentially closer to the concept of correlation or relation, since it is simply derived from how well the other variables fit the target variable."**
>
> A3: In FRF, we describe the relationship between nodes with functions, and they can be treated as equality constraints, f(x) = 0 or x_i = g(x_{\backslash i}), derived from how well the other variables fit the target variable. Thus, these learned functional relations are precise and explicit, not just simply correlation. An obvious advantage of this precise functional relationship once learned could be directly applied to test phase as a strong inductive bias. Therefore, this distinguishes from traditional graph models only consisting of connected nodes.
>
>
> **Q4: "This is highly data or observation-dependent."**
> A4: It is true that the functional relations to be learned are data-dependent, which is exactly the idea of the FRF. Employing FRF, in the synthetic and real-world applications we considered, we found the learned functional relations could successfully recover the inter-node relationship. For example, in binary tree case, the underlying relationship between nodes are faithfully recovered through learning from data.
>
>
> **Q5: "Moreover, it needs a threshold to determine the set of relevant variables. For multi-variables in different value domains or distributions, finding proper thresholds seems nontrivial and would affect the overall performance."**
>
> A5: When training the constraint network (i.e. functional relation network), the validation loss will be very small if the constraint relation exists (for miniapp calling applications). Therefore, it is is easy to choose a threshold to filter the constraint nodes using both training and validation data. And the proposed model is insensitive to the threshold. Empirically, it is effective to set the error threshold in range of 0.001 - 0.025. In Figure 3, the threshold is setting to a small value 0.005 since the Binary tree dataset is clean data. Meanwhile, the error threshold is setting to a slightly larger 0.025 to achieve the best performance because there are noise in traffic dataset and the validation loss is bigger than that in binary tree dataset.

---

> ### Comment · Reviewer_RLjD · 2022-12-05
> **Score maintained**
>
> Thanks for the authors' reply.
>
> The methodology of identifying constraints simply based on function fitting is still unconvincing. The predictive relation between multivariates does not necessarily correspond to equal constraints that need to hold in the training and inference. Though the author claims some conceptual advantages of this way of identifying constraints, essentially what the proposed method does is just the variable-wise neural network fitting to the rest of the variables, and it is not directly translated to equal constraints among variables.
>
> The method of determining the set of relevant variables in a constraint seems arbitrary depending on the data-specific thresholding, and the authors' reply gives no systematic or principled methods to handle this.
>
> Meanwhile, the technical implementation of the proposed method is standard and shows little innovation, neither a novel combination of existing techniques nor new insights.
>
> Overall, the score would be maintained at 3.

---

> > ### Author Response · Authors · 2022-12-05
> > **Thanks and some answers to your comments**
> >
> > **Q1: The predictive relation between multivariates does not necessarily correspond to equal constraints that need to hold in the training and inference**
> >
> > A1:  In the training and inference stages, the targets are the same time-series in different intervals, $x_0, ..., x_t-1$ in training and $x_t, ..., x_{t+\tau}$ in inference. Therefore, we think the same time-series should maintain the same function relation. In many real-world applications, it is make sense, such as the web-page calling relation and the road network in traffic flow forecasting which is rarely changed at most time. In this paper, we consider the static function relation, so the relation should be mantained in training and inference stage.
> >
> > **Q2: the proposed method does is just the variable-wise neural network fitting to the rest of the variables.**
> >
> > A2: The method in this paper is the first attempt to solve the multi-variate time-series forecasting as a **non-linear constrained convex optimization problem**, not just the variable-wise neural network fitting to the rest of the variable. The constraint network is a simple but effective part to find the relevant nodes, the most important part in this paper is **the applying of the constraint function to the forecasting** shown in eq.13.
> >
> >
> > **Q3:The method of determining the set of relevant variables in a constraint seems arbitrary depending on the data-specific thresholding, and the authors' reply gives no systematic or principled methods to handle this.**
> >
> > A3: The simple 2-layer network is efficient to capture the complex relation between multi-variate time-series. As shown in Figure 4, the binary tree structure is well discovered by the constraint network. Meanwhile, both Figure.3 and Figure.5 show the performance of FRF is not sensitive to the hyper-parameters
> >
> >
> > **Q4:  the technical implementation of the proposed method is standard and shows little innovation, neither a novel combination of existing techniques nor new insights.**
> >
> > A4: There are two differences between FRF and existing methods.
> >
> > First, most multi-variate time-series forecasting is under a single framework, such as STGCN, AGCRN and SCINet which is model-specific. However, FRF is model-agnostic and can be applied to transformer families, GCN families and DNN families, and improve these backbones significantly and consistently.
> >
> > Second, existing methods modeling multi-variate relationship based on the assumption that the flow prediction of the target node may benifit from the simliar flows, while we are the first focus on the constraint function rather than the similarity relation.

---

> > > ### Comment · Reviewer_RLjD · 2022-12-06
> > > **Reply**
> > >
> > >
> > > Forecasting models have residual distributions, which could differ between training and testing, and significantly change if distribution shift is taken into account. Given the residual distribution with tails, equating predictive relations to equal constraints is not intuitively sound, not to mention the need for the proper ways of adapting constraints to time series data where heterogeneity commonly exists.
> > >
> > > If the constraints are not well defined, formulating the constrained optimization problem makes little sense in my opinion.
> > >
> > > As for Q3, I am afraid that is a repeated argument that does not address the issue.
> > >
> > > As for Q4, given the aforementioned fundamental flaws, being model-agnostic would be just a side advantage.

---

> > > > ### Author Response · Authors · 2022-12-06
> > > > **Some answers to your concerns-Part2**
> > > >
> > > > Q4: As for Q3, I am afraid that is a repeated argument that does not address the issue.
> > > >
> > > > A4: I think the data-specific threshold is not a disadvantage of this paper, the important point is the sensitivity of hyper-parameters. As demonstrated in experiments, the model is insensitive to hyper-parameters.
> > > >
> > > > Q5: As for Q4, given the aforementioned fundamental flaws, being model-agnostic would be just a side advantage.
> > > >
> > > > A5: The performances of model-specific multi-variate time-series models under fixed paradigms (GCN families, RNN families) may be failed in some cases such as tail distribution. However, the model-agnostic function relation framework can benefit from the progress of the uni-variate forecasting model, then improve the performance consistently. In experiments, we attempt to add the FRF to the SOTA uni-variate models (FEFformer, Autoformer, SCInet), FRF can enhance most SOAT models including multi-variate and uni-variate models.
> > > >
> > > > Therefore, the model-agnostic framework is important for multi-variate forecasting.

---

> > > > ### Author Response · Authors · 2022-12-06
> > > > **Some answers to your concerns-Part1**
> > > >
> > > > Q1: Forecasting models have residual distributions, which could differ between training and testing, and significantly change if distribution shift is taken into account.
> > > >
> > > > A1:
> > > > If we consider uni-variate time series prediction problem, it is true that it is  difficult to predict the future when the distribution of testing data is significantly different from the training distribution.
> > > >
> > > > However, the topic we are studying in this work is multi-variate time series. In this scenario, despite that the change of uni-variate data distribution is very common in real-world applications, among the multiple time series there are still some inherent functional relations that are maintained between training and test data. It is indeed worthy of  exploring these relations between these time series and applying these relations to improve prediction performance. This is exactly what have done in this work and its effectiveness is soundly verified through extensive empirical results.
> > > >
> > > > In a word, the distributional difference between training and test for one single time series  does not mean that the relationship between multi-variate time series is different.
> > > >
> > > > Q2: Given the residual distribution with tails, equating predictive relations to equal constraints is not intuitively sound, not to mention the need for the proper ways of adapting constraints to time series data where heterogeneity commonly exists.
> > > >
> > > > A2: We answer this question from the following three aspects.
> > > >
> > > >  First, in a real-world application, the time series has three inherent attributions: closeness, seasonality, and tendency. If a time series is only an unknown tail distribution and has no seasonality, so the model should focus on the tendency. I think how to model the distribution of a uni-variate time series whether tail distribution or seasonality is not the main topic of this paper, we focus on modeling the function relation of multi-variate time series, then applying the relation to improve the performance.
> > > >
> > > > In other words, **a uni-variate  time series which has tendency or seasonality doesn't impact the multi-variate relation essentially. Considering the multi-variate relation should be from a global view rather than a uni-variate  distribution view**. The relation should be well discovered and applied to forecasting which is the purpose of most multi-variate forecasting works. Moreover, the inherent function relation between multi-variate in real-world applications is maintained in most scenarios. For example, in the mini-app calling flow, web calling flow, or the QPS in large-scale cloud-native service, the relation between the mini-app or cloud machine is maintained in most cases, which only changes some nodes in one or two years.
> > > >
> > > > Overall, how to model the uni-variate  time-series distribution such as tail distribution, residual distribution or seasonal distribution is another topic.  And, how to model the function relation between multi-variate and then apply the relation  is the point of this paper.
> > > >
> > > > More importantly, the proposed FRF can be applied to many uni-variate time-series frameworks such as FEDformer, Autoformer, SCInet, and improve uni-variate time-series performance consistently, which is demonstrated in our experiments.
> > > >
> > > > Second, for real-world applications, such as cloud-native systems, traffic systems and mini-app management systems, the flow of each node usually has closeness, seasonality, and tendency. In addition, the relation between multi-variate is time-invariant.
> > > >
> > > > Third, even if the data distribution of the target time series is the long tail, the function relation is still maintained when the calling relation is unchanged and the up-down streams are also long tails.
> > > >
> > > >
> > > > Q3: If the constraints are not well defined, formulating the constrained optimization problem makes little sense in my opinion.
> > > >
> > > > A3: We answer this question from the following two aspects.
> > > >
> > > > First, The constraint can be well discovered by a simple but effective constraint network in many real-world applications. As shown in figure 4, the binary tree structure is recovered by the 2-layer attention network. In addition, the mini-app calling constraints and traffic constraints are both well-discovered, and are demonstrated in experiments.
> > > >
> > > > Second, the perspective of modeling multi-variate time-series forecasting as a constrained optimization problem is make sense when the constraint is static or dynamic. Even if the static  constraint is not well defined in some applications, the dynamic constraints can also follow the constrained optimization framework.

---

> > > > > ### Comment · Reviewer_RLjD · 2022-12-06
> > > > > **Reply to authors arguments**
> > > > >
> > > > > I am afraid the above argument did not address the key point.
> > > > >
> > > > > For any regression/forecasting model, there is a residual distribution that reflects the probability characteristics of the data, model quality, etc. The proposed method amounts to getting rid of residuals by imposing equal constraints. I am afraid I still do not see a reasonable theoretical intuition behind it. In practice, this method will bias the training phase toward the high-density area of the residual distribution and overlook the tail behaviors of the target, while in many applications the model's performance on the tail matters.

---

> > > > > > ### Author Response · Authors · 2022-12-07
> > > > > > **Emphasis on the central theme we have investigated**
> > > > > >
> > > > > > We acknowledge that how to deal with distributional shift between training and test data is an important aspect for any prediction model in machine learning. However, we emphasize again that **our research goal in this work is to investigate the functional relations between multiple time series and how to leverage these relations to improve the prediction performance of original prediction models**. **How to handle the  distributional difference between training and test for a single time series is out of scope of our study**, which could be a future direction.
> > > > > >
> > > > > > It is widely accpeted that a single paper cannot successfully solve all the challenging issues in a large research area. In the decades of research on time series prediction, many key problems still remain and continually challenge academics and industrial research, such as interpretability, distributional shift, how to model multi-variate time series, how to make prediction for very long horizon, etc. Our work mainly focuses on how to leverage the relationship between multiple time series to further improve prediciton performance.

---

> > > > > > > ### Comment · Reviewer_RLjD · 2022-12-07
> > > > > > > **Unrelated argument**
> > > > > > >
> > > > > > > Eq.(1) to Eq.(4) formulate the forecasting problem as a probability model. My last comment has nothing to do with the distribution shift and it is about some basic rationale behind probability models that the authors seem to be unfamiliar with. Thus, the argument is self-repeated and not on the same page.
> > > > > > >
> > > > > > > Meanwhile, given the probability model formation in Eq.(1) to Eq.(4), the proposed method forces predictions to follow deterministic constraints instead. This is a self-contradictory design and oversimplifies the probabilistic characteristic of forecasting problems.

---

> > > > > > > > ### Author Response · Authors · 2022-12-07
> > > > > > > > **Clarification about the probability model and loss function**
> > > > > > > >
> > > > > > > > Sorry for the misunderstanding on your previous argument on distribution of the probability model. Now we make clarification about this point.
> > > > > > > >
> > > > > > > > You are right that Eq.(1) to Eq.(4) formulate the forecasting problem as a probability model. This a general formuation on the time series prediction, which is used in many works such as [1]. Actually, we typically assume the probability model $P(\{y_{t+1},...,y_{t+M}\} | \{x_{t- H+1},...,x_t\})$ is a **Gaussian likelihood**, with its mean as the output of the backbone prediction network for each time step, \hat{y}_{t+1},...,\hat{y}_{t+M} $= G_{\theta}(\{x_{t- H+1},...,x_t\})$. So the final loss function $L(\hat{y}, y)$ is reduced to standard mean square error (MSE) for each predicted time step. We will clarify this point in the new version. Sorry for the confusion.
> > > > > > > >
> > > > > > > > [1] Bing Yu, Haoteng Yin, and Zhanxing Zhu. Spatio-temporal graph convolutional networks: A deep learning framework for traffic forecasting. In IJCAI, 2018.

---

### Official Review · Reviewer_dWww · 2022-11-05

**Confidence:** 4
**Correctness:** 4
**Technical Novelty And Significance:** 3
**Empirical Novelty And Significance:** 3
**Recommendation:** 6

**Clarity, Quality, Novelty And Reproducibility:**

The paper is written well and provides source code in the supplementary for reproducibility.

**Strength And Weaknesses:**

Strength:

The proposed method is novel. There are no existing papers that consider the complicated constraints relationship among nodes.

The proposed method is technically sound

Empirical performance is promising

Weakness:

The paper did not compare with some baselines that perform joint forecasting and structure learning, e.g., GTS [1] and NIR [2].

[1] DISCRETE GRAPH STRUCTURE LEARNING FOR FORECASTING MULTIPLE TIME SERIES, ICLR 2021
[2] Neural Relational Inference for Interacting Systems. ICML 2018


**Summary Of The Paper:**

This paper is about multivariate time series forecasting with structure learning. Existing works usually assume the graph structure of the multiple time series is given or learned by
the node similarity. However, in some applications, the relationship between time series can be much more complicated and the graph structure is not enough. This paper proposes to
use functional relation field to model the inter-node relationship. Experiments on one synthetic and two real-world datasets show that the proposed method can enhance the existing
spatial-temporal forecasting model.

**Summary Of The Review:**

The paper proposes a novel method to learn the complicated structure for multi-variate time series forecasting task. However, it misses some important baseline.

---

> ### Author Response · Authors · 2022-11-19
> **Thank you for recognizing the novelty of our approach and we add more experiments in the revised verison.**
>
> In the following, we take "Q" as the question from reviewers and "A" as our answer.
>
> **Q1: The paper did not compare with some baselines that perform joint forecasting and structure learning, e.g., GTS [1] and NIR [2].**
>
> **A1:  We conduct more experiments on GTS [1] and NRI [2] in Table.5 in the Appendix of update version.**
>
> In  order to demonstrate that our proposed FRF can consistently improve the backbone prediction models,
> we also incorporate the FRF into two more approaches involved with graph structure learning, GTS
> (Shang et al., 2021) and NRI (Kipf et al., 2018).
>
> The first one GTS learns the graph structure among multiple time series and forecasts them simultaneously with a GNN. There are two differences between GTS and our proposed FRF. On one hand, GTS performs prediction under GNN paradigm which is model-specific while FRF is model-agnostic applying the functional relation field to forecasting loss optimization. On the other hand, existing stud ies including AGCRN and GTS construct the graph based on the similarity between time series, while the FRF is the first approach that proposes to exploit the the constraint function relation to enhance the multi-variate time series forecasting. We conduct experiments on Binary tree, MiniApp1 and MiniApp2 datasets using the opensourced code (https://github.com/chaoshangcs/GTS.git) shown in Table.5, demonstrating that **FRF can also improve the forecasting performance on GTS**. The code and running logs of FRF enhanced GTS model is released in the Supplementary Material.
>
> The second model neural relational inference (NRI) is an unsupervised one that learns to infer interactions and forecasting with a LSTM. We conduct experiments on Binary tree, Miniapp1 and Miniapp2 dataset using the opensourced code (https://github.com/ethanfetaya/NRI.git). The results on NRI network in Table.5. The MAE on these three dataset on FRF-NRI are $19.61$, $2.47$, $7.98$, while NRI is $22.77$, $2.50$, $8.04$ on three dataset repspectively. We can easily observe that **FRF can consistently improve the prediction performance of NRI for these datasets**. We will release the code and running logs of FRF enhanced NRI model in the future.
>
> [1] DISCRETE GRAPH STRUCTURE LEARNING FOR FORECASTING MULTIPLE TIME SERIES, ICLR 2021
>
> [2] Neural Relational Inference for Interacting Systems. ICML 2018

---

> > ### Comment · Reviewer_dWww · 2022-12-06
> > **Retain the Score**
> >
> > I would like to thank the authors for the added experiments. The authors have addressed my concerns during the rebuttal. The original score (6) has already been left some room for the improvement, so won't be further increased.

---

### Author Response · Authors · 2022-11-19
**General reply to all the reviewers**

We thank all reviewers' recognition on the novelty of our paper and efforts for providing instructive feedback.

According to the reviews, we still would like to highlight two major advantages of the proposed framework for multi-variate times series prediction: it is **model-agonistic** and can be incorporated into any backbone prediction network (including those involving graph structure learning ); and the introduced inductive bias can **consistently improve the prediction performance of backbone networks**. We first learn the functional relationship between nodes. This could be thought as an inductive bias that will be incorporated to training and test phases of the backbone network in the form of regularization. Thus, we emphasize that FRF is **not aiming to learn graph structure**, and **the recovered graph structure by FRF is only a by-product** of our framework.

We have replied to all the reviewers’ questions as below and updated the paper, particularly adding the experimental comparison with more methods as required by the reviewers and adding the discussions on the hyper-parameters and computational complexity. All these are included in the Appendix of the updated version.

---

### Decision · Program_Chairs · 2023-01-20

**Decision:**

Reject

**Justification For Why Not Higher Score:**

The paper has many unresolved issues as pointed out in the reviews and in the above meta review. Also the formulation and approaches used in each step are questionable.

**Justification For Why Not Lower Score:**

N/A

**Metareview: Summary, Strengths And Weaknesses:**

The paper proposes to model inter-variate constraints in multivariate time series and imposing these constraints on the predicted output during training and inference. The approach is demonstrated on a toy problem and two real-world dataset.

Strength: The paper consider an important problem, namely leveraging the relationship among time series to improve multivariate forecasting. The exposition is clear and the two-stage approach interesting

Weaknesses: Unfortunately, several concerns remain about the present submission, and we strongly encourage the authors to pursue their work to address them in depth.
- In particular, the proposed process that identifies the relevant node for each target node individually might be suboptimal compared to joint identification for all nodes (similarly to graphical lasso vs neighborhood selection in gaussian graphical modeling ). Also selection of relevant nodes based on gradients is questionable and prone the identification of spurious dependencies. Important dependency might also me missed due to the use of a single threshold across nodes.  A two-staged process seems suboptimal compared to a single stage process where sparsity in imposed, e.g. using sparse attention networks.
- Regularizing the prediction loss with the constrain satisfaction squared loss is quite ad-hoc as argued by Reviewer dWww.
- Also it seems suboptimal that the learning of the functional relational field should be done once and for all, and then this fixed field should be applied.
- From a formulation perspective, step (a) in Figure 2 of the paper ignores what can be predicted using past values of all the time series, while it might be argued that one might be better off if one considers that the functional relationship might be akin to "instantaneous dependencies" and should be estimated after subtracting what can be accounted for / predicted  by the past values of the times series.
- As pointed out by Reviewer n4W8 , the number of parameters is quite daunting. Also heir tuning and sensitivity should be studied more systematically.